# On the Entropy and the Maximum Entropy Principle of Uncertain Variables

**DOI:** 10.3390/e25081195

**Published:** 2023-08-11

**Authors:** Yujun Liu, Guanzhong Ma

**Affiliations:** School of Mathematics and Statistics, Anyang Normal University, Anyang 455000, China; liuyj@aynu.edu.cn

**Keywords:** entropy, maximum entropy principle, uncertainty theory, generalized inverse of an increasing function, substitution rule for Lebesgue–Stieltjes integrals

## Abstract

A new variance formula is developed using the generalized inverse of an increasing function. Based on the variance formula, a new entropy formula for any uncertain variable is provided. Most of the entropy formulas in the literature are special cases of the new entropy formula. Using the new entropy formula, the maximum entropy distribution for unimodel entropy of uncertain variables is provided without using the Euler–Lagrange equation.

## 1. Introduction

The concept of entropy was first introduced by Rudolf Clausius in the mid-19th century as a measure of the disorder or randomness in a system. In 1948, Shannon [1] inaugurated information entropy in information theory, which measures the unpredictability of the realization of a random variable. Subsequently, information entropy developed into an essential tool in information theory (see Gray [2]). Entropy was introduced into dynamical systems by Kolmogorov [3] in 1958 to investigate the conjugacy problems of the Bernoulli shift. In 2009, Liu [4] proposed logarithmic entropy in uncertainty theory to quantify the difficulty in forecasting the outcome of an uncertain variable. In 2012, Knuth et al. [5] gave an interesting application of maximal joint entropy in automated learning machines. Currently, entropy is widely used in various fields, such as probability theory, ergodic theory, dynamical systems, machine learning and data analysis, uncertainty theory, and others. We refer the readers to Martin and England [6], Walters [7], Einsiedler and Thomas [8], and Liu [9] for a detailed introduction to the entropy theory in these fields.

The maximum entropy principle was first expounded by Jaynes [10,11], where he gave a natural criterion of choice, i.e., from the set of all probability distributions compatible with several mean values of several random variables, choose the one that maximizes Shannon entropy. From then on, the methods of maximum entropy became widely applied tools for constructing the probability distribution in statistical inference [12], decision-making [13,14], communication theory [15], time-series analysis [16], and reliability theory [17,18]. In 2017, Abbas et al. [19] made an elaborate comparison between the methods of maximum entropy and the maximum log-probability from the point of the Kullback–Leibler divergence. In this paper, we will discuss the entropy and the maximum entropy principle in uncertainty theory.

Uncertainty theory, founded by Liu [20] in 2007 and refined in 2015 [9], is a branch of axiomatic mathematics that deals with human’s belief degree based on normality, duality, sub-additivity, and product axioms. Within the past few years, it has attracted numerous researchers. Peng and Iwamura [21], Liu and Lio [22] provided a sufficient and necessary characterization of an uncertainty distribution. Based on the inverse uncertainty distribution, Yao [23] introduced a new approach to computing the variance. Subsequently, Sheng and Kar [24] extended Yao’s result to the moments of the uncertain variable.

After the introduction of entropy in uncertainty theory, the entropy and maximum entropy principle were researched by numerous experts. Inspired by the maximum entropy principles in information theory and statistics, Chen and Dai [25] explored the maximum entropy principle of logarithmic entropy. According to the maximum entropy principle, using the Lagrange multipliers method, Chen and Dai [25] proved that normal uncertainty distribution is the maximal entropy distribution under the prescribed constraints on the expected value and variance. In addition, using Fubini’s theorem, Dai and Chen [26] built an entropy formula for regular uncertain variables (see Definition 2). Subsequently, there have been many extensions of entropy, the entropy formula, and the corresponding maximum entropy distribution. Yao et al. [27] proposed sine entropy. Yao et al. [27] provided an entropy formula for regular uncertain variables, and the authors also obtained the maximum entropy distribution for sine entropy. Tang and Gao [28] proposed triangular entropy and provided an entropy formula for regular uncertain variables. Ning et al. [29] applied triangular entropy to portfolio selection. Subsequently, Dai [30] suggested quadratic entropy and provided an entropy formula for regular uncertain variables. In addition, Dai [30] found the maximum entropy distribution for quadratic entropy. Here, we point out, all the entropy formulas in Dai and Chen [26], Yao et al. [27], Tang and Gao [28], and Dai [30] are based on Fubini’s theorem. Subsequently, Ma [31] proposed unimodal entropy of uncertain variables and provided an entropy formula for regular uncertain variables. Ma [31] also found the maximum entropy distribution for unimodal entropy for regular uncertain variables. Nevertheless, limited by their methods, all the entropy formulas in Dai and Chen [26], Yao et al. [27], Tang and Gao [28], Dai [30], and Ma [31] are only established for regular uncertain variables. We know most uncertain variables are not regular. Therefore, it has some significance not only in theory but also in practice to prove the entropy formulas of Dai and Chen [26], Yao et al. [27], Tang and Gao [28], Dai [30], and Ma [31] are still true for any uncertain variable. In this paper, we will provide an entropy formula that is true for any uncertain variable by using the generalized inverse of an increasing function. We emphasize that the entropy formulas of Dai and Chen [26], Yao et al. [27], Tang and Gao [28], Dai [30], and Ma [31] are special cases of our entropy formula, and our method differs completely from those methods used by Dai and Chen [26], Yao et al. [27], Dai [30], and Ma [31].

Furthermore, we point out that all the maximum entropy distributions of Chen and Dai [25], Yao et al. [27], and Dai [30] were obtained based on the Lagrange multipliers method. Using our entropy formula, we will obtain the maximum entropy distribution without utilizing the Lagrange multipliers method. We also emphasize that the maximum entropy distributions of Chen and Dai [25], Yao et al. [27], and Dai [30] are special cases of ours.

The rest of the paper is arranged as follows. In Section 2, we first collect some necessary facts in uncertainty theory and some requisite results of the generalized inverse of an increasing function. Then, we review the entropy, entropy formulas, and the maximum entropy principles in uncertainty theory. In Section 3, we present our entropy formula and our maximum entropy distribution. We also demonstrate some applications of our entropy formula and maximum entropy distribution. In Section 4, we prove our entropy formula and derive the maximum entropy distribution of unimodel entropy. In Section 5, we verify that the maximum entropy distributions of logarithmic entropy in Theorem 2, sine entropy in Theorem 5, and quadratic entropy in Theorem 8 are special cases of our maximum entropy distributions. Finally, a concise discussion and further research are given in Section 6.

## 2. Preliminaries

In this section, first, we will collect the necessary facts in uncertainty theory. Second, we will recall some requisite results of the generalized inverse of an increasing function. Finally, we review the entropy, entropy formulas, and the maximum entropy principles in uncertainty theory.

### 2.1. Some Primary Facts in Uncertainty Theory

Let Γ be a non-empty set and suppose L is a σ-algebra over Γ. Each element *A* in L is called an event. To handle belief degree reasonably, Liu [20] introduced the uncertain measure.

**Definition** **1.**
*A function M:L→[0,1] is called an uncertain measure provided M satisfies the following three axioms:*
1.
*Normality: M{Γ}=1;*
2.
*Duality: $M\left\{A\right\}+M\left\{A^{c}\right\}=1$ for each event A;*
3.
*Subadditivity: for each sequence of events $A_{1},A_{2},\ldots$, one gets*

$$M\left\{\overset{\infty }{\underset{i=1}{⋃}}A_{i}\right\}\leq \sum_{i=1}^{\infty }M\left\{A_{i}\right\}.$$




The triplet (Γ,L,M) is called an uncertainty space. A measurable function X:(Γ,L,M)→R is called an uncertain variable. Let *X* be an uncertain variable; the function Φ(x)=M{X≤x} is called the uncertainty distribution of *X*.

**Definition** **2**([32])**.**
*An uncertainty distribution Φ is called regular provided it is strictly increasing and continuous over {x:0<Φ(x)<1}, and*
$$\lim_{x\to -\infty }\Phi \left(x\right)=0,\;\lim_{x\to +\infty }\Phi \left(x\right)=1.$$
*An uncertain variable is called regular if its uncertainty distribution is regular.*

Obviously, a regular uncertainty distribution Φ is invertible. Now, we recall two regular uncertainty distributions that are most used in uncertainty theory. The first is the normal uncertainty distribution N(e,σ) [32], which is defined by
(1)$$\Phi \left(x\right)={\left(1+\exp\left(\frac{\pi (e-x)}{\sqrt{3}\sigma }\right)\right)}^{-1},\;x\in R.$$
Here, *e* and σ are real numbers with σ>0. We call an uncertain variable *X* normal provided its uncertainty distribution is normal and write X∼N(e,σ) (Figure 1).

The second is the linear uncertainty distribution L(a,b) [32], which is defined by
$$\Phi \left(x\right)=\left\{\begin{array}{cc} 0, & x\leq a\\ \frac{x-a}{b-a}, & 0<x\leq b\\ 1, & b<x \end{array}\right.$$
where *a* and *b* are real numbers with a<b (Figure 2).

Liu [20] suggested the expectation to describe the mean of an uncertain variable.

**Definition** **3**([20])**.**
*Let X be an uncertain variable. The expectation of X is defined by*
$$E\left[X\right]=\int_{0}^{\infty }M\left\{X\geq x\right\}dx-\int_{-\infty }^{0}M\left\{X\leq x\right\}dx$$
*if one or two of the integrals above are finite.*

Let *X* be an uncertain variable with finite expectation *e*; the variance of *X* is defined by
$$V\left[X\right]=E[{\left(X-e\right)}^{2}].$$
Given an uncertainty variable *X* with uncertainty distribution Φ and finite expectation *e*, the variance of *X* can be computed by (see Section 2.6 of Chapter 2 in [9])
(2)$$V\left[X\right]=\int_{0}^{+\infty }\left(1-\Phi \left(e+\sqrt{x}\right)+\Phi \left(e-\sqrt{x}\right)\right)dx.$$

Yao [23] provided a nice formula for calculating the variance by inverse uncertainty distribution.

**Theorem** **1**(Yao [23])**.**
*Assume that X is a regular uncertain variable with uncertainty distribution Φ. If E[X] is finite, then*
$$V\left[X\right]=\int_{0}^{1}{\left(\Phi^{-1}\left(\alpha \right)-E\left[X\right]\right)}^{2}d\alpha .$$

We will prove the variance formula of Yao is true for any uncertain variable in Theorem 9, which is one of our main tools for establishing our entropy formula and maximum entropy principle.

### 2.2. Necessary Results of the Generalized Inverse of an Increasing Function

To present our results, we need to review the generalized inverse of an increasing function, which is a basic tool in statistics; see Embrechts and Hofert [33], Resnick [34], Kampke-Radermacher [35]. Here, we follow the definition of Embrechts and Hofert [33].

**Definition** **4**([33])**.**
*Let Φ:R→[0,1] be an increasing function. The generalized inverse of Φ is defined by*
(3)$$\Phi^{-}\left(\alpha \right)=\inf\left\{x\in R:\Phi \left(x\right)\geq \alpha \right\},\;\alpha \in \left(0,1\right],$$
*with the convention that inf{∅}=+∞, where ∅ is the empty set.*
*Specifically, we define $\Phi^{-}\left(0\right)=\sup\left\{x\in R:\Phi \left(x\right)=0\right\}$ with the convention that sup{∅}=−∞.*


**Remark** **1.**
*To stress the difference between the generalized inverse and standard inverse, we denote the generalized inverse of an increasing function Φ by $\Phi^{-}$. If Φ is strictly increasing and continuous, $\Phi^{-}$ is just the standard inverse of Φ.*


We also point out that, by the above definition, $\Phi^{-}\left(0\right)$ may be −∞ and $\Phi^{-}\left(1\right)$ may be +*∞*. Here is an example used frequently in uncertainty theory.

**Example** **1.**
*Let Φ be a normal uncertainty distribution defined by (Equation 1). Then $\Phi^{-}$ is:*

$$\Phi^{-}\left(\alpha \right)=\left\{\begin{array}{cc} -\infty , & \alpha =0\\ e-\frac{\sqrt{3}\sigma }{\pi }\ln\frac{1-\alpha }{\alpha }, & 0<\alpha <1\\ +\infty , & \alpha =1. \end{array}\right.$$



We also need to introduce the generalized inverse of a left-continuous increasing function; see Appendix B in Iritani and Kuga [36].

**Definition** **5**([36])**.**
*Let G be a left-continuous and increasing function over (0,1). The generalized inverse of G is defined by*
(4)$$G^{-1}\left(x\right)=\sup\left\{y:G(y)\leq x\right\}$$
*for all real numbers x.*

The succeeding interesting fact is that the generalized inverse of the generalized inverse for a right-continuous increasing function is just itself. To the best of our knowledge, this result was proved first by Iritani and Kuga [36] and was refined by Kampke-Radermacher [35].

**Lemma** **1**([35]). *Let Φ:R→[0,1] be a right-continuous and increasing function; G is the generalized inverse of Φ defined by (Equation 3), i.e., $G=\Phi^{-}$. Then*
$$G^{-1}\left(x\right)=\Phi \left(x\right)$$
*for all x∈R, where $G^{-1}\left(x\right)$ is the generalized inverse of G defined by (Equation 4).*

### 2.3. Entropy Formula and the Principle of Maximum Entropy

To quantify the difficulty in predicting the outcomes of an uncertain variable, Liu [4] proposed logarithm entropy.

**Definition** **6**([4]). *Let X be an uncertain variable with uncertainty distribution Φ. The logarithm entropy of X is defined by*
(5)$$H_{L}\left[X\right]=\int_{-\infty }^{\infty }L\left(\Phi (x)\right)dx.$$
*Here, L(y)=−ylny−(1−y)ln(1−y) for y∈(0,1) and L(0)=L(1)=0.*

Using the Lagrange multipliers method, Chen and Dai [25] obtained the maximum entropy distribution for the logarithmic entropy.

**Theorem** **2**([25])**.**
*Suppose X is an uncertain variable with finite expectation e and variance $\sigma^{2}$. Then,*
(6)$$H_{L}\left[X\right]\leq \frac{\sigma \pi }{\sqrt{3}},$$
*and the equality holds if and only if X is a normal uncertain variable N(e,σ).*

In addition, using Fubini’s theorem, Dai and Chen [26] attained an interesting entropy formula for regular uncertain variables.

**Theorem** **3**([26])**.**
*Suppose X is a regular uncertain variable with uncertainty distribution Φ. If $H_{L}\left[X\right]$ is finite, we have*
(7)$$H_{L}\left[X\right]=\int_{0}^{1}\Phi^{-1}\left(\alpha \right)\ln\frac{\alpha }{1-\alpha }d\;\alpha .$$

Subsequently, Yao et al. [27] proposed sine entropy.

**Definition** **7**([27]). *Let X be an uncertain variable with uncertainty distribution Φ. The sine entropy of X is defined by*
(8)$$H_{S}\left[X\right]=\int_{-\infty }^{\infty }S\left(\Phi (x)\right)dx.$$
*Here, S(y)=sin(πy) for y∈[0,1].*

Yao et al. [27] provided an entropy formula and obtained the maximum entropy distribution of sine entropy.

**Theorem** **4**([27])**.**
*Suppose X is a regular uncertain variable with uncertainty distribution Φ. Then, we have*
(9)$$H_{S}\left[X\right]=\pi \int_{0}^{1}\Phi^{-1}\left(1-\alpha \right)\cos\left(\pi \alpha \right)d\alpha .$$

**Theorem** **5**([27])**.**
*Suppose X is an uncertain variable with uncertainty distribution Φ, expectation e, and variance $\sigma^{2}$. Then, we have*
(10)$$H_{S}\left[X\right]\leq \frac{\sqrt{2}}{2}\pi \sigma$$
*and equality holds if and only if*
$$\Phi^{-1}\left(\alpha \right)=e-\sigma \sqrt{2}\cos\left(\pi \alpha \right),\;\alpha \in \left(0,1\right).$$

Tang and Gao [28] suggested triangular entropy and found an entropy formula.

**Definition** **8**([28]). *Let X be an uncertain variable with uncertainty distribution Φ. The triangular entropy of X is defined by*
(11)$$H_{T}\left[X\right]=\int_{-\infty }^{\infty }T\left(\Phi (x)\right)dx,$$
*where*
$$T\left(y\right)=\left\{\begin{array}{cc} y, & 0\leq y\leq 0.5,\\ 1-y, & 0.5\leq y\leq 1. \end{array}\right.$$

**Theorem** **6**([28])**.**
*Suppose X is a regular uncertain variable with uncertainty distribution Φ. Then, we have*
$$H_{T}\left[X\right]=-\int_{0}^{0.5}\Phi^{-1}\left(\alpha \right)d\alpha +\int_{0.5}^{1}\Phi^{-1}\left(\alpha \right)d\alpha .$$

Dai [30] put forward quadratic entropy.

**Definition** **9**([30]). *Let X be an uncertain variable with uncertainty distribution Φ. The quadratic entropy of X is defined by*
(12)$$H_{Q}\left[X\right]=\int_{-\infty }^{\infty }Q\left(\Phi (x)\right)dx.$$
*Here, Q(y)=y(1−y) for y∈[0,1].*

Dai [30] also provided an entropy formula and got the maximum entropy distribution of quadratic entropy.

**Theorem** **7**([30])**.**
*Suppose X is a regular uncertain variable with uncertainty distribution Φ. If the entropy exists, then*
(13)$$H_{Q}\left[X\right]=\int_{0}^{1}\Phi^{-1}\left(\alpha \right)\left(2\alpha -1\right)d\alpha .$$

**Theorem** **8**([30])**.**
*Suppose X is an uncertain variable with uncertainty distribution Φ, expectation e, and variance $\sigma^{2}$. Then, we have*
(14)$$H_{Q}\left[X\right]\leq \frac{\sigma }{\sqrt{3}}$$
*and equality holds if and only if*
$$\Phi^{-1}\left(\alpha \right)=e+\sigma \sqrt{3}\left(2\alpha -1\right),\;\alpha \in \left(0,1\right).$$

Ma [31] suggested unimodal entropy.

**Definition** **10**([31]). *Let X be an uncertain variable with uncertainty distribution Φ. The unimodal entropy of X is defined by*
(15)$$H_{U}\left[X\right]=\int_{-\infty }^{\infty }U\left(\Phi (x)\right)dx,$$
*where function U is increasing on (0,0.5] and decreasing on [0.5,1), U(0)=U(1)=0.*

Ma [31] also provided an entropy formula and got the maximum entropy distribution of the unimodal entropy for regular uncertain variables. The readers can refer to Liu [9], Yao et al. [27], Tang and Gao [28], Dai [30], Ma [31], and the references therein for details.

Here, we emphasize the entropy formulas of Dai and Chen [26], Yao et al. [27], Tang and Gao [28], Dai [30], and Ma [31] hold only for regular uncertain variables. We will develop an entropy formula that is true for any uncertain variable. To establish the entropy formulas, the method used by Dai and Chen [26], Yao et al. [27], Tang and Gao [28], and Dai [30] is Fubini’s theorem. We stress that Fubini’s theorem does not work in our case because the uncertainty distribution Φ may not be regular here. We need to develop new methods.

Finally, we point out, to obtain the maximum entropy distribution, the main method used by Chen and Dai [25], Yao et al. [27], and Dai [30] is the Lagrange multipliers method. Based on our entropy formula, we can obtain the maximum entropy distribution without using the Lagrange multipliers method. We also stress that the only tool we need is the Cauchy–Schwartz inequality. In the following section, we will exhibit our new entropy formula and provide the maximum entropy distribution of unimodel entropy.

## 3. Results

In this section, we will present our results and give some remarks. First, we point out that the variance formula of Yao [23] for regular uncertain variables is valid for any uncertain variable.

**Theorem** **9.**
*Let X be an uncertain variable with uncertainty distribution Φ, finite expectation e, and finite variance. Then*

$$V\left[X\right]=\int_{0}^{1}{\left(\Phi^{-}\left(\alpha \right)-e\right)}^{2}d\alpha ,$$

*where $\Phi^{-}$ is the generalized inverse of Φ.*


We leave the verification of Theorem 9 to the next section. Based on our variance formula and some facts on the generalized inverse, we will develop an entropy formula for any uncertain variable.

**Theorem** **10.**
*Let X be an uncertain variable with finite expectation and finite variance. Assume further that unimodal function U is differentiable and satisfies*

(16)
$$\int_{0}^{1}{\left|U^{\prime }\left(\alpha \right)\right|}^{2}d\alpha <\infty .$$

*Then we have*

(17)
$$H_{U}\left[X\right]=-\int_{0}^{1}\Phi^{-}\left(\alpha \right)U^{{}^{\prime }}\left(\alpha \right)d\;\alpha ,$$

*where $\Phi^{-}$ is the generalized inverse of the uncertainty distribution Φ of X.*


**Remark** **2.**
1.
*The entropy formulas in Dai and Chen [26], Yao et al. [27], Tang and Gao [28], Dai [30], and Ma [31] are valid only for regular uncertain variables. Our entropy formula holds for any uncertain variable.*
2.
*Fubini’s theorem, which is used by Dai and Chen [26], Yao et al. [27], Tang and Gao [28], and Dai [30] to establish the entropy formula there, does not work in our setting, because the uncertainty distribution is not necessarily regular here.*



We specifically point out, under the assumption that the uncertainty distribution Φ is regular, let U(α)=−αlnα−(1−α)ln(1−α), U(α)=sin(πt), U(α)=α(1−α),
$$U\left(\alpha \right)=\left\{\begin{array}{cc} \alpha , & 0\leq \alpha \leq 0.5,\\ 1-\alpha , & 0.5\leq \alpha \leq 1 \end{array}\right.$$
respectively. By Theorem 10, we can regain the entropy formulas of logarithm entropy in Theorem 3, sine entropy in Theorem 4, triangular entropy in Theorem 6, and quadratic entropy in Theorem 7. As the verifications are routine, we omit them.

Using our entropy formula in Theorem 10, we obtain the maximum entropy distribution of unimodal entropy.

**Theorem** **11.**
*Let X be an uncertain variable with uncertainty distribution Φ, finite expectation e, and finite variance $\sigma^{2}$. Assume further that unimodal function U is differentiable and satisfies*

(18)
$$\int_{0}^{1}{\left|U^{\prime }\left(\alpha \right)\right|}^{2}d\alpha <\infty .$$

*Then, the unimodal entropy of X satisfies*

(19)
$$H_{U}\left[X\right]\leq \sigma \sqrt{\int_{0}^{1}{\left|U^{\prime }\left(\alpha \right)\right|}^{2}d\alpha }$$

*and the equality holds if and only if*

(20)
$$\Phi^{-1}\left(\alpha \right)=e-cU^{\prime }\left(\alpha \right),\;\alpha \in \left(0,1\right)$$

*where c is a real number such that*

(21)
$$\left|c\right|\sqrt{\int_{0}^{1}{\left|U^{\prime }\left(\alpha \right)\right|}^{2}d\alpha }=\sigma .$$



**Remark** **3.**
*We stress that the maximum entropy distribution of the unimodal entropy of Ma [31] is provided only for regular uncertain variables. Here, our maximum entropy distribution of unimodal entropy in Theorem 11 is obtained for any uncertain variable.*


As applications of our maximum entropy distribution in Theorem 11, we can deduce the maximum entropy distribution of logarithm entropy in Theorem 2, sine entropy in Theorem 5, and quadratic entropy in Theorem 8 immediately.

**Proposition** **1**(Chen and Dai [25])**.**
*Let X be an uncertain variable with uncertainty distribution Φ, finite expectation e, and variance $\sigma^{2}$. Then, the logarithmic entropy of X satisfies*
$$H_{L}\left[X\right]\leq \frac{\pi \sigma }{\sqrt{3}}$$
*and the equality holds if and only if*
$$\Phi^{-1}\left(\alpha \right)=e+\frac{\sigma \sqrt{3}}{\pi }\ln\frac{\alpha }{1-\alpha },\;\alpha \in \left(0,1\right),$$
*i.e., X is a normal uncertain variable N(e,σ).*

**Proposition** **2**(Yao et al. [27])**.**
*Let X be an uncertain variable with uncertainty distribution Φ, finite expectation e, and variance $\sigma^{2}$. Then, its sine entropy satisfies*
$$H_{S}\left[X\right]\leq \frac{\pi \sigma }{\sqrt{2}}$$
*and the equality holds if and only if*
$$\Phi^{-1}\left(\alpha \right)=e-\sigma \sqrt{2}\cos\left(\pi \alpha \right),\;\alpha \in \left(0,1\right).$$

**Proposition** **3**(Dai [30])**.**
*Let X be an uncertain variable with uncertainty distribution Φ, finite expectation e, and variance $\sigma^{2}$. Then, its quadratic entropy satisfies*
$$H_{Q}\left[X\right]\leq \frac{\sigma }{\sqrt{3}}$$
*and the equality holds if and only if*
$$\Phi^{-1}\left(\alpha \right)=e+\sigma \sqrt{3}\left(2\alpha -1\right),\;\alpha \in \left(0,1\right).$$

We leave the proofs of Corollary 1, Corollary 2, and Corollary 3 to Section 6. In the following section, we will prove Theorem 9, Theorem 10, and Theorem 11.

## 4. Proof of Theorem 9, Theorem 10, and Theorem 11

To prove Theorems 9 and 10, we need a result of the substitution rule on Lebesgue–Stieltjes integrals, which is a conclusion of Proposition 2 of Falkner and Teschl [37].

**Lemma** **2**([37])**.**
*Assume function M is increasing over [a,b] and function N is left-continuous increasing over [M(a),M(b)]. Then, for each bounded Borel function f over [a,b], one has*
(22)$$\int_{a}^{b}f\left(x\right)dN\left(M\left(x\right)\right)=\int_{M(a)}^{M(b)}f\left(M^{-1}\left(y\right)\right)dN\left(y\right),$$
*where $M^{-1}$ is the generalized inverse of M defined by (Equation 4).*

Here, we point out that if N(y)=y, this result goes back to the classical case of Lebesgue [38]. Before proving the main results, we also need a simple fact.

**Lemma** **3.**
*Suppose function g is increasing on (0,1) with*

$$\lim_{x\to 0^{+}}g\left(x\right)=-\infty \;and\;\lim_{x\to 1^{-}}g\left(x\right)=+\infty ,$$

*and let L be a function increasing on (0,a] and decreasing on [a,1), respectively, for some a in (0,1). Then*
(a)
*$|\int_{0}^{a}g\left(x\right)dL\left(x\right)|<+\infty$ means*

$$\lim_{A\to 0^{+}}g\left(A\right)\left(L\left(A\right)-L\left(0\right)\right)=0;$$

(b)
*$|\int_{a}^{1}g\left(x\right)dL\left(x\right)|<+\infty$ means*

$$\lim_{B\to 1^{-}}g\left(B\right)\left(L\left(1\right)-L\left(B\right)\right)=0.$$




Because the verification of Lemma 3 is a routine computation, we omit it.

We also point out that, notwithstanding that uncertainty distribution Φ may not be right-continuous, since Φ is increasing, the assumption that Φ is right-continuous does not change the variance and the entropy of an uncertain variable. Thus, we assume uncertainty distribution Φ is right-continuous in the rest. Now, we will prove Theorem 9.

**Proof** **of** **Theorem** **9**.Let Y=X−e and Ψ be the uncertainty distribution of *Y*. Then we have
$$E\left[Y\right]=0,V\left[X\right]=V\left[Y\right],\;\mathrm{and}\;\Psi^{-}=\Phi^{-}-e.$$We only need to prove
$$V\left[Y\right]=\int_{0}^{1}{\left(\Psi^{-}\left(\alpha \right)\right)}^{2}d\alpha .$$Recall (see equality (Equation 2))
$$V\left[Y\right]=\int_{0}^{\infty }\left(1-\Psi \left(\sqrt{x}\right)\right)dx+\int_{0}^{\infty }\Psi \left(-\sqrt{x}\right)dx.$$
We only verify
$$\int_{0}^{\infty }\left(1-\Psi \left(\sqrt{x}\right)\right)dx=\int_{\Psi (0)}^{1}{\left(\Psi^{-}\left(\alpha \right)\right)}^{2}d\alpha .$$
The verification of
$$\int_{0}^{\infty }\Psi \left(-\sqrt{x}\right)dx=\int_{0}^{\Psi (0)}{\left(\Psi^{-}\left(\alpha \right)\right)}^{2}d\alpha$$
is similar. First, note that
$$\int_{0}^{\infty }\left(1-\Psi \left(\sqrt{x}\right)\right)dx=\lim_{B\to 1^{-}}\int_{0}^{\Psi^{-}\left(B\right)}\left(1-\Psi \left(\sqrt{x}\right)\right)dx.$$Letting $y=\sqrt{x}$, we have
$$\begin{array}{cc} \int_{0}^{\Psi^{-}\left(B\right)}\left(1-\Psi \left(\sqrt{x}\right)\right)dx & =\int_{0}^{\Psi^{-}\left(B\right)}\left(1-\Psi \left(y\right)\right)dy^{2}\\ \; & =\int_{0}^{\Psi^{-}\left(\Psi \left(0\right)\right)}\left(1-\Psi \left(y\right)\right)dy^{2}+\int_{\Psi^{-}\left(\Psi \left(0\right)\right)}^{\Psi^{-}\left(B\right)}\left(1-\Psi \left(y\right)\right)dy^{2}\\ \; & =\left(1-\Psi \left(0\right)\right){\left(\Psi^{-}\left(\Psi \left(0\right)\right)\right)}^{2}+\int_{\Psi^{-}\left(\Psi \left(0\right)\right)}^{\Psi^{-}\left(B\right)}\left(1-\Psi \left(y\right)\right)dy^{2}, \end{array}$$
where the last equality is because Ψ(y)≡Ψ(0) for $y\in \left(\Psi^{-}\left(\Psi \left(0\right)\right),0\right)$.By Lemma 2, let $M=\Psi^{-}$, f(x)=1−x, $N\left(y\right)=y^{2}$, noting that $M^{-}=\Psi$ according to Lemma 1, we have
$$\int_{\Psi^{-}\left(\Psi \left(0\right)\right)}^{\Psi^{-}\left(B\right)}\left(1-\Psi \left(y\right)\right)dy^{2}=\int_{\Psi (0)}^{B}\left(1-x\right)d{\left(\Psi^{-}\left(x\right)\right)}^{2}.$$
By integration by parts, we have
$$\int_{\Psi (0)}^{B}\left(1-x\right)d{\left(\Psi^{-}\left(x\right)\right)}^{2}=\left(1-B\right){\left(\Psi^{-}\left(B\right)\right)}^{2}-\left(1-\Psi \left(0\right)\right){\left(\Psi^{-}\left(\Psi \left(0\right)\right)\right)}^{2}+\int_{\Psi (0)}^{B}{\left(\Psi^{-}\left(x\right)\right)}^{2}dx.$$
Then, we have
$$\int_{0}^{\Psi^{-}\left(B\right)}\left(1-\Psi \left(\sqrt{x}\right)\right)dx=\left(1-B\right){\left(\Psi^{-}\left(B\right)\right)}^{2}+\int_{\Psi (0)}^{B}{\left(\Psi^{-}\left(x\right)\right)}^{2}dx.$$
Let
$$G\left(B\right)=\int_{\Psi (0)}^{B}{\left(\Psi^{-}\left(x\right)\right)}^{2}dx.$$
Note that function *G* is increasing over [Ψ(0),1]. It follows that
$$\lim_{B\to 1^{-}}G\left(B\right)=\lim_{n\to +\infty }G\left(1-\frac{1}{n}\right).$$
By monotone convergence theorem, we have
$$\begin{array}{cc} \int_{\Psi (0)}^{1}{\left(\Psi^{-}\left(x\right)\right)}^{2}dx & =\lim_{n\to +\infty }\int_{\Psi (0)}^{1}{\left(\Psi^{-}\left(x\right)\right)}^{2}1_{\left[\Psi \left(0\right),1-\frac{1}{n}\right]}\left(x\right)dx\\ \; & =\lim_{n\to +\infty }G\left(1-\frac{1}{n}\right), \end{array}$$
where $1_{\left[\Psi \left(0\right),1-\frac{1}{n}\right]}$ is indicator function of the interval $\left[\Psi \left(0\right),1-\frac{1}{n}\right]$.Thus, we have
$$\lim_{B\to 1^{-}}\int_{\Psi (0)}^{B}{\left(\Psi^{-}\left(x\right)\right)}^{2}dx=\int_{\Psi (0)}^{1}{\left(\Psi^{-}\left(x\right)\right)}^{2}dx.$$
We need to prove
$$\lim_{B\to 1^{-}}\left(1-B\right){\left(\Psi^{-}\left(B\right)\right)}^{2}=0.$$
If
$$\lim_{B\to 1^{-}}\Psi^{-}\left(B\right)<+\infty ,$$
the result is obvious. If
$$\lim_{B\to 1^{-}}\Psi^{-}\left(B\right)=+\infty ,$$
by Lemma 3, let $g\left(x\right)={\left(\Psi^{-}\left(x\right)\right)}^{2}$ and L(x)=x, we have
$$\lim_{B\to 1^{-}}\left(1-B\right){\left(\Psi^{-}\left(B\right)\right)}^{2}=0.$$
Thus, we finish the proof of Theorem 9.  □

Now, using Theorem 9, Lemma 2, and Lemma 3, we will prove Theorem 10, i.e., the entropy formula.

**Proof** **of** **Theorem** **10**.Recall $$H_{U}\left[X\right]=\int_{-\infty }^{\infty }U\left(\Phi (x)\right)dx.$$Note that
$$\Phi \left(x\right)=\left\{\begin{array}{cc} 0, & x\leq \Phi^{-}\left(0\right),\\ 1, & x\geq \Phi^{-}\left(1\right) \end{array}\right.$$
and U(0)=U(1)=0. We have
$$\int_{-\infty }^{\infty }U\left(\Phi (x)\right)dx=\int_{\Phi^{-}\left(0\right)}^{\Phi^{-}\left(1\right)}U\left(\Phi (x)\right)dx.$$By the monotone convergence theorem, we have
$$\int_{\Phi^{-}\left(0\right)}^{\Phi^{-}\left(1\right)}U\left(\Phi (x)\right)dx=\lim_{n\to +\infty }\int_{\Phi^{-}\left(0\right)}^{\Phi^{-}\left(1\right)}U\left(\Phi (x)\right)1_{\left[\Phi^{-}\left(\frac{1}{n}\right),\Phi^{-}\left(1-\frac{1}{n}\right)\right]}\left(x\right)dx,$$
where
$$1_{\left[\Phi^{-}\left(\frac{1}{n}\right),\Phi^{-}\left(1-\frac{1}{n}\right)\right]}\left(x\right)$$
is indicator function of the interval $\left[\Phi^{-}\left(\frac{1}{n}\right),\Phi^{-}\left(1-\frac{1}{n}\right)\right]$.Let
$$G\left(\epsilon \right)=\lim_{\epsilon \to 0^{+}}\int_{\Phi^{-}\left(0\right)}^{\Phi^{-}\left(1\right)}U\left(\Phi (x)\right)1_{\left[\Phi^{-}\left(\epsilon \right),\Phi^{-}\left(1-\epsilon \right)\right]}\left(x\right)dx.$$
Note that G(ϵ) is decreasing over [0,1]. We have
$$\lim_{n\to +\infty }G\left(\frac{1}{n}\right)=\lim_{\epsilon \to 0^{+}}G\left(\epsilon \right).$$Thus
$$\begin{array}{cc} \int_{\Phi^{-}\left(0\right)}^{\Phi^{-}\left(1\right)}U\left(\Phi (x)\right)dx & =\lim_{n\to +\infty }G\left(\frac{1}{n}\right)\\ \; & =\lim_{\epsilon \to 0^{+}}\int_{\Phi^{-}\left(\epsilon \right)}^{\Phi^{-}\left(1-\epsilon \right)}U\left(\Phi \left(x\right)\right)dx. \end{array}$$Let f=U, $M=\Phi^{-}$, N(x)=x; by Lemma 2 and integration by parts, we have
$$\begin{array}{cc} \int_{\Phi^{-}\left(\epsilon \right)}^{\Phi^{-}\left(1-\epsilon \right)}U\left(\Phi \left(x\right)\right)dx & =\int_{\epsilon }^{1-\epsilon }U\left(\alpha \right)d\Phi^{-}\left(\alpha \right)\\ \; & =U\left(1-\epsilon \right)\Phi^{-}\left(1-\epsilon \right)-U\left(\epsilon \right)\Phi^{-}\left(\epsilon \right)-\int_{\epsilon }^{1-\epsilon }\Phi^{-}\left(\alpha \right))dU\left(\alpha \right). \end{array}$$
Then, we have
$$H_{U}\left(X\right)=\lim_{\epsilon \to 0^{+}}U\left(1-\epsilon \right)\Phi^{-}\left(1-\epsilon \right)-\lim_{\epsilon \to 0^{+}}U\left(\epsilon \right)\Phi^{-}\left(\epsilon \right)-\lim_{\epsilon \to 0^{+}}\int_{\epsilon }^{1-\epsilon }\Phi^{-}\left(\alpha \right)dU\left(\alpha \right).$$
We only need to show
$$\lim_{\epsilon \to 0^{+}}U\left(1-\epsilon \right)\Phi^{-}\left(1-\epsilon \right)=0,\lim_{\epsilon \to 0^{+}}U\left(\epsilon \right)\Phi^{-}\left(\epsilon \right)=0$$
and
$$\lim_{\epsilon \to 0^{+}}\int_{\epsilon }^{1-\epsilon }\Phi^{-}\left(\alpha \right)dU\left(\alpha \right)=\int_{0}^{1}\Phi^{-}\left(\alpha \right)U^{{}^{\prime }}\left(\alpha \right)d\alpha .$$
Noting that $\int_{0}^{1}U^{{}^{\prime }}\left(\alpha \right)d\alpha =0$, by the Cauchy–Schwartz inequality and Theorem 9, we have
$$\begin{array}{cc} \left|\int_{0}^{1}\Phi^{-}\left(\alpha \right)dU\left(\alpha \right)\right| & =\left|\int_{0}^{1}\left(\Phi^{-}\left(\alpha \right)-e\right)U^{{}^{\prime }}\left(\alpha \right)d\alpha \right|\\ \; & \leq \sqrt{\int_{0}^{1}{\left(\Phi^{-}\left(\alpha \right)-e\right)}^{2}d\alpha }\sqrt{\int_{0}^{1}{\left(U^{{}^{\prime }}\left(\alpha \right)\right)}^{2}d\alpha }\\ \; & =\sigma \sqrt{\int_{0}^{1}{\left(U^{{}^{\prime }}\left(\alpha \right)\right)}^{2}d\alpha }<+\infty . \end{array}$$
By monotone convergence theorem, again we have
$$\lim_{\epsilon \to 0^{+}}\int_{\epsilon }^{1-\epsilon }\Phi^{-}\left(\alpha \right)dU\left(\alpha \right)=\int_{0}^{1}\Phi^{-}\left(\alpha \right)U^{{}^{\prime }}\left(\alpha \right)d\alpha .$$
Let L=U and $g=\Phi^{-}$; by Lemma 3, we have
$$\lim_{\epsilon \to 0^{+}}U\left(1-\epsilon \right)\Phi^{-}\left(1-\epsilon \right)=0\;\;\mathrm{and}\;\lim_{\epsilon \to 0^{+}}U\left(\epsilon \right)\Phi^{-}\left(\epsilon \right)=0.$$
Then, we have
$$H_{U}\left[X\right]=-\int_{0}^{1}\Phi^{-}\left(\alpha \right)U^{{}^{\prime }}\left(\alpha \right)d\;\alpha .$$
Thus, we finish the proof of Theorem 10.  □

Now, based on our entropy formula, we can derive the maximum entropy distribution of unimodal entropy immediately.

**Proof** **of** **Theorem** **11**.Noting that $\int_{0}^{1}U^{{}^{\prime }}\left(\alpha \right)d\alpha =0$, by Theorem 10, we have
$$H_{U}\left[X\right]=-\int_{0}^{1}\left(\Phi^{-}\left(\alpha \right)-e\right)U^{{}^{\prime }}\left(\alpha \right)d\;\alpha .$$
By the Cauchy–Schwartz inequality and Theorem 9, we have
$$\begin{array}{cc} H_{U}\left[X\right] & \leq \sqrt{\int_{0}^{1}{\left(\Phi^{-}\left(\alpha \right)-e\right)}^{2}d\alpha }\sqrt{\int_{0}^{1}{\left(U^{{}^{\prime }}\left(\alpha \right)\right)}^{2}d\alpha }\\ \; & =\sigma \sqrt{\int_{0}^{1}{\left(U^{{}^{\prime }}\left(\alpha \right)\right)}^{2}d\alpha }, \end{array}$$
and “=” holds if and only if there exists some real number *c* such that
$$\Phi^{-}\left(\alpha \right)-e=cU^{{}^{\prime }}\left(\alpha \right).$$
Then, we have
$$\int_{0}^{1}{\left(\Phi^{-}\left(\alpha \right)-e\right)}^{2}d\alpha =c^{2}\int_{0}^{1}{\left(U^{{}^{\prime }}\left(\alpha \right)\right)}^{2}d\alpha ,$$
i.e.,
$$\left|c\right|\sqrt{\int_{0}^{1}{\left|U^{\prime }\left(\alpha \right)\right|}^{2}d\alpha }=\sigma .$$
Thus, Theorem 11 is proved.  □

## 5. Proof of Proposition 1, Proposition 2, and Proposition 3

In this section, we use our maximum entropy distribution to deduce Proposition 1, Proposition 2, and Proposition 3, i.e., the maximum entropy distributions of logarithm entropy, sine entropy, and quadratic entropy. As the proofs of Proposition 2 and Proposition 3 are similar to that of Proposition 1, we only prove Proposition 1.

**Proof** **of** **Proposition** **1**.Let U(α)=−αlnα−(1−α)ln(1−α),α∈(0,1).
Then,
$$U^{{}^{\prime }}\left(\alpha \right)=\ln\frac{1-\alpha }{\alpha }.$$
By Theorem 2 of Ma et al. [39], we have
$$\int_{0}^{1}{\left(U^{{}^{\prime }}\left(\alpha \right)\right)}^{2}d\alpha =\frac{\pi^{2}}{3}.$$
By Theorem 11, we have
$$H_{L}\left[X\right]\leq \frac{\pi \sigma }{\sqrt{3}}$$
and the equality holds if and only if
(23)$$\Phi^{-1}\left(\alpha \right)-e=c\ln\frac{\alpha }{1-\alpha },\;\alpha \in \left(0,1\right).$$
Since functions
$$\Phi^{-1}\left(\alpha \right)\;\mathrm{and}\;\ln\frac{\alpha }{1-\alpha }$$
are increasing for α∈(0,1), the real number *c* is positive. By equality (Equation 23), we have
$$\int_{0}^{1}{\left(\Phi^{-1}\left(\alpha \right)-e\right)}^{2}d\alpha =c^{2}\int_{0}^{1}{\left(\ln\frac{\alpha }{1-\alpha }\right)}^{2}d\alpha .$$
Thus
$$c=\frac{\sigma \sqrt{3}}{\pi }.$$
Then
$$\Phi^{-1}\left(\alpha \right)=e+\frac{\sigma \sqrt{3}}{\pi }\ln\frac{\alpha }{1-\alpha },\;\alpha \in \left(0,1\right),$$
i.e., Φ is a normal uncertainty distribution with expectation *e* and variance $\sigma^{2}$.  □

## 6. Discussion

In this paper, by using the generalized inverse of an increasing function, we developed a useful entropy formula that is valid for any uncertain variable and encompasses all the previous cases in the literature. In addition, based on our entropy formula, we obtained the maximum entropy distribution for unimodel entropy. We also point out that most of the entropy formulas and maximum entropy distributions of the uncertain variables in the literature are special cases of ours.

Further research can consider the following two directions:(1)Develop the entropy formula and the maximum entropy principle of an uncertain set;(2)Establish the entropy formula and research the maximum entropy distributions of uncertain random variables.

## Figures and Tables

**Figure 1 entropy-25-01195-f001:**
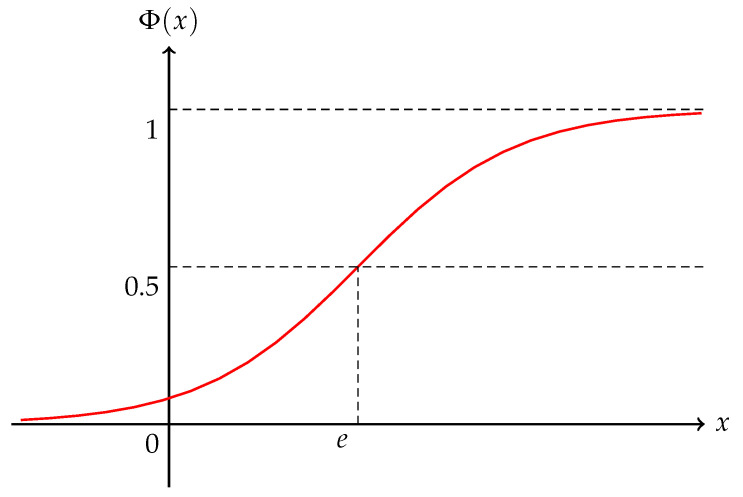
Normal uncertainty distribution N(e,σ).

**Figure 2 entropy-25-01195-f002:**
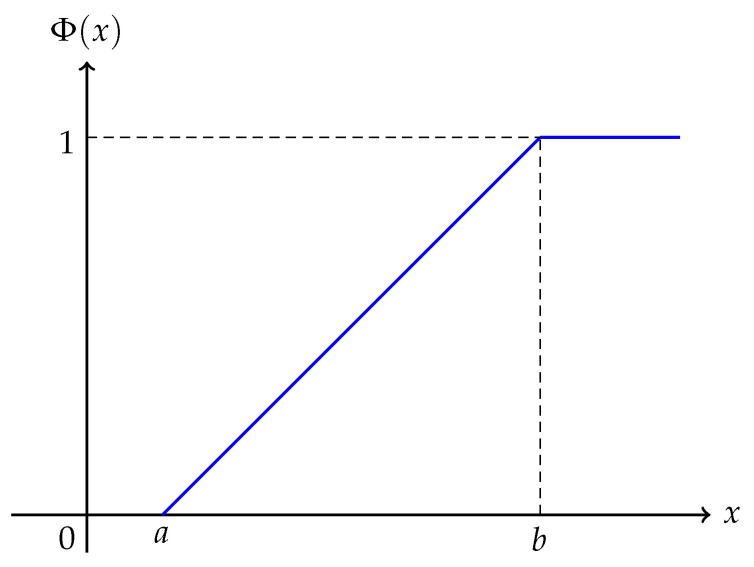
Linear uncertainty distribution L(a,b).

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
