# Peer review of "On the Entropy and the Maximum Entropy Principle of Uncertain Variables"

_entropy, 2023, doi:10.3390/e25081195_

Round 1

Reviewer 1 Report

Using generalized inverse of increasing function, the authors developed an entropy formula for uncertain variable. They established a new maximum entropy principle for uncertain random variable.

For journal publication of this paper, the authors solve the following problems.

(1) The authors have to explain their research results and future works in conclusion section. But, I did not find it. So, they have to create a conclusion section and add the conclusions on their researches and experimental to this section.

(2) The Preliminaries section, the authors showed so many definitions and theorems. I recommend that the authors shorten this section to include only the essentials needed for the proposed researches.

(3) If possible, the authors are requested to add simulation study to support their proposed theory. This result is expected to be necessary for a detailed understanding of this paper.

Reviewer 2 Report

This is an interesting article.  However, I had difficulty reviewing it as it is written for an audience that is much more advanced than me in applied mathematics.

I would have liked to see some examples and plots of typical distributions and their entropy differences with the new relationship to the variance.

I cannot judge the article's impact on a unified description formula without some results and plots allowing me to asses the correctness.  

Reviewer 3 Report

please see the attached pdf file

The quality of English need is good but sometime the construction of the phrase is wrong. As an example:

- line 149 "the succeeding interesting fact" is wrong

-line 164 "if the Euler-Lagrange equation can solve..." should be " if the following maximization problem can be solved using "

Round 2

Reviewer 2 Report

Definitely, the manuscript has improved.  I am satisfied that the authors have done their best and although I am unable to apply this to some of the practical data sets I work with I am happy to recommend publication.

Reviewer 3 Report

The authors have addressed my remarks and now the paper Is improved and fit for publication in m'y opinion

Minor  english checking Is required